# Case Report: A Rare Manifestation of Pulmonary Arterial Hypertension in Ankylosing Spondylitis

**DOI:** 10.3390/jpm13010062

**Published:** 2022-12-28

**Authors:** Tsung-Yuan Yang, Yong-Hsin Chen, Wun-Zhih Siao, Gwo-Ping Jong

**Affiliations:** 1Division of Cardiology, Department of Internal Medicine, Chung Shan Medical University Hospital, Taichung 40201, Taiwan; 2Institute of Medicine, College of Medicine, Chung Shan Medical University, Taichung 40201, Taiwan; 3Department of Occupational Safety and Health, Chung Shan Medical University Hospital, Chung Shan Medical University, Taichung 40201, Taiwan

**Keywords:** ankylosing spondylitis, pulmonary arterial hypertension, extra-articular presentation, connective tissue disease, rare manifestation

## Abstract

Ankylosing spondylitis (AS) is an inflammatory disease that involves the axial skeleton and can present with extra-articular manifestations. However, there are scarce reports describing the link between AS and pulmonary arterial hypertension (PAH). Herein, we report on a 58-year-old man with a history of AS for 32 years who developed PAH as confirmed by echocardiography and right cardiac catheterization. To our knowledge, this is the first case of AS associated with PAH 32 years after the AS diagnosis with a detailed clinical description. We are recommended that physicians should be aware of this rare comorbidity in AS patients. Early echocardiographic screening is necessary for symptomatic patients.

## 1. Background

An extra-articular presentation is common in patients with AS, including uveitis, inflammatory bowel disease, and psoriasis [1]. Cardiovascular involvement has also been described in AS, manifested as aortitis, cardiac conduction disturbances, and valvular heart disease [2]. Pulmonary manifestations in AS, such as interstitial lung disease, apical fibrobullous disease, obstructive sleep apnea, and spontaneous pneumothorax, are less frequent [3]. However, very few reports describe the link between AS and pulmonary arterial hypertension (PAH). Herein, we report on a middle-aged man with a history of young-aged AS for 32 years associated with PAH.

## 2. Case Presentation

A 58-year-old man had a history of AS with stable disease activity following administration of the disease-modifying antirheumatic drug sulfasalazine at a dose of 5 mg twice per day for 32 years. He complained of progressive exertional dyspnea and leg edema and felt shortness of breath for the last 3 months after climbing two floors of stairs. He was admitted to our hospital under the impression of PAH and the World Health Organization’s (WHO) functional class IV [4]. This patient was never a smoker and denied any history of exposure to toxic chemicals.

Split-second pulmonary heart sounds with bilateral lower extremity edema were discovered during the examination. Laboratory testing (white blood cell, hemoglobin, platelet, and liver function tests) results were within normal limits, except for blood urea nitrogen of 27 mg/dL (reference range <20 mg/dL), creatinine of 1.42 mg/dL (reference range <1.30 mg/dL), high-sensitivity troponin I of 48.8 pg/mL(reference range <17.5 pg/mL), and N-terminal pro-brain natriuretic peptide (NT-pro-BNP) of 4528 pg/mL (reference range <300 pg/mL) (Table 1). The 6 min walk distance (6MWD) was 340 m, desaturation from 98% to 82%. Grade 3 bilateral sacroiliitis was diagnosed according to the modified New York classification on a plain radiograph examination of the pelvis (Figure 1) [5]. A chest film demonstrated enlarged bilateral main pulmonary arteries with abrupt tapering of the peripheral pulmonary vasculature (Figure 2). The electrocardiograph disclosed an inverted T-wave in the precordial leads, and the inferior leads II, III, aVF, and aVF suggested right ventricular strain. Echocardiography revealed preserved left ventricular systolic function (left ventricular ejection fraction = 59.7%). No significant valvular heart disease was detected except for severe tricuspid regurgitation. The estimated systolic pulmonary artery pressure was approximately 96 mmHg. Dilated right ventricle (RV), a high RV index of myocardial performance (RIMP, 0.64; reference range <0.54) (Figure 3A) and low fractional area change (FAC) of the RV (21%; reference range >35%) (Figure 3B) suggested impaired global right ventricular function (Table 2). A D-shaped left ventricle evidenced during systole and diastole can be the result of high right ventricular pressure and volume overload (Figure 4A,B). No intracardiac shunt finding was detected.

Thyroid function was normal, and a test for the human immunodeficiency virus was negative. Pulmonary function testing revealed normal but mild impairment of the diffusing capacity of the lung for carbon monoxide (DLCO), which was 15.6% of the predicted value. Abdominal sonography conducted negative findings for portal hypertension or cirrhosis. No evidence of pulmonary embolism, pulmonary emphysema, vasculitis, associated interstitial lung disease, or granulomatous infection was observed according to a high-resolution computed tomography scan. A right cardiac catheterization study demonstrated main pulmonary artery pressure of 85/23 mmHg (mean = 43 mmHg), RV pressure of 84/11 mmHg, mean right atrial pressure of 18 mmHg, pulmonary artery wedge pressure of 18 mmHg, pulmonary vascular resistance of 9.39 WU, and a cardiac index of 2.11 L/min/m^2^. No definite causes for the development of PAH were disclosed in this case. Hence, the patient was diagnosed with AS associated with PAH.

During hospitalization, the patient was treated with diuretics and was discharged with symptomatic improvement on the 6MWD (430 m) and the NT-pro-BNP level (1171 pg/mL). During the outpatient clinic follow-up, 62.5 mg of bosentan twice per day was initiated. Then, the prescription was switched to 20 mg of sildenafil three times per day following approval from the National Health Insurance Bureau. One month later, 10 mg of macitentan every day was added to the sildenafil because the patient still suffered from dyspnea (Table 3). The NT-pro-BNP level decreased from 1171 to 98 pg/mL, and the 6MWD increased from 430 to 550 m 3 months after starting the macitentan treatment. The patient was under outpatient departmental follow-up in stable condition.

## 3. Discussion

To our knowledge, this is the first case of AS associated with PAH 32 years after the AS diagnosis. The diagnosis of PAH was confirmed by transthoracic echocardiography and right heart catheterization.

Autoimmune diseases such as systemic sclerosis, systemic lupus erythematosus, and mixed connective tissue disease are common etiologies of PAH [6]. However, only a few case reports have described the association between AS and PAH. Karoli et al. illustrated the high incidence of pulmonary hypertension in AS patients [7]. However, their study made the diagnosis of PAH by echocardiography without right heart catheterization data. The coexistence of another connective tissue disease was not excluded, which may have contributed to pulmonary hypertension.

In contrast, Hung et al. described a rare case of a 27-year-old man with a 12-year history of AS who developed PAH. The diagnosis was confirmed by echocardiography and right heart catheterization [8]. Another observational study reported that older age, a long smoking history, AS duration, and poor spinal mobility increase the risk of PAH in patients with AS [9]. Endothelial dysfunction may play an essential role in pathogenesis [9].

In the present case, we reported a middle-aged man with AS associated with PAH 32 years after the AS diagnosis. The abnormal RIMP and RV FAC indicated RV dysfunction, which may have been caused by pulmonary hypertension. The diagnoses of thyroid disease, HIV-associated PAH, cirrhosis, portal hypertension, lung disease, left heart failure, and pulmonary emboli were excluded because of negative laboratory findings. The high pulmonary capillary wedge pressure suggested that the elevated left ventricular end-diastolic pressure may result from fluid overload related to kidney injury. Left-sided valvular heart disease is another etiology of PAH [10,11], and we excluded left-sided valvular heart disease in our case by echocardiography. Thus, PAH was diagnosed as associated with AS.

In summary, we report an AS patient with PAH. An echocardiogram is suggested for all patients with symptomatic who have heart failure. Physicians should be aware of this rare comorbidity in AS patients. Early screening is necessary for symptomatic patients using modalities such as echocardiography. Further epidemiologic studies are needed to disclose the association between AS and PAH.

## Figures and Tables

**Figure 1 jpm-13-00062-f001:**
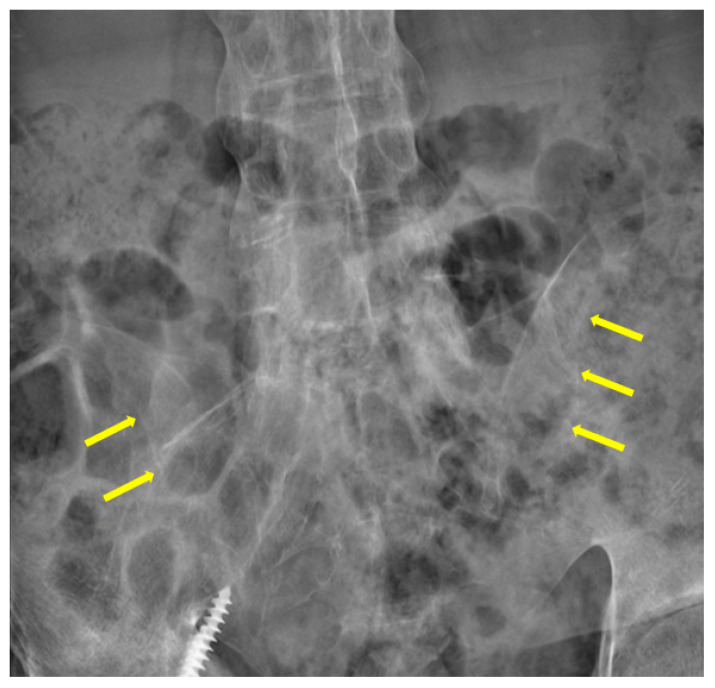
A plain radiograph of the pelvis demonstrated sclerosis, partial ankylosis and narrowing (yellowish arrows) over the sacroiliac joints, suggesting grade-3 bilateral sacroiliitis.

**Figure 2 jpm-13-00062-f002:**
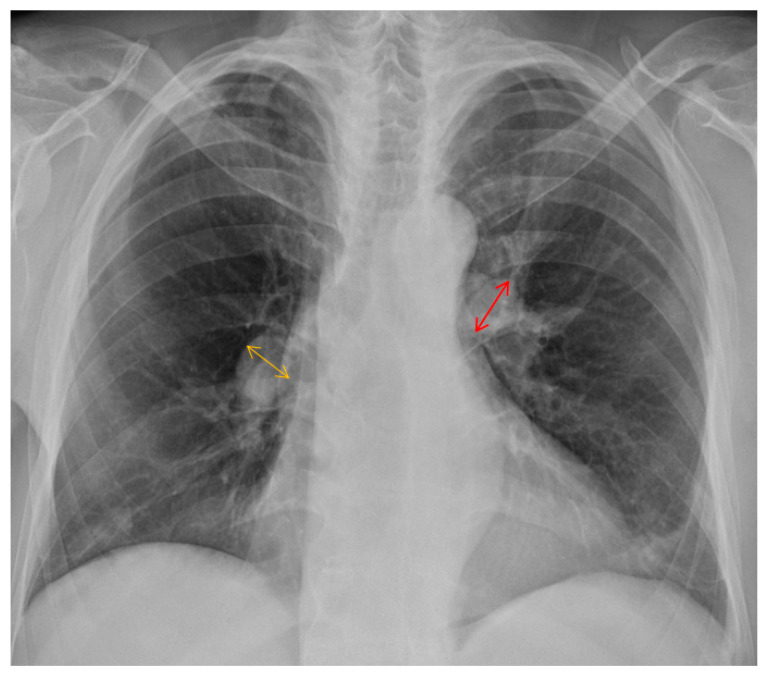
Chest radiograph showed engorged bilateral main pulmonary arteries and pruning of peripheral pulmonary vessels. The orange and reddish lines indicate the diameter of the right (21 mm) and left (24 mm) main pulmonary artery, respectively.

**Figure 3 jpm-13-00062-f003:**
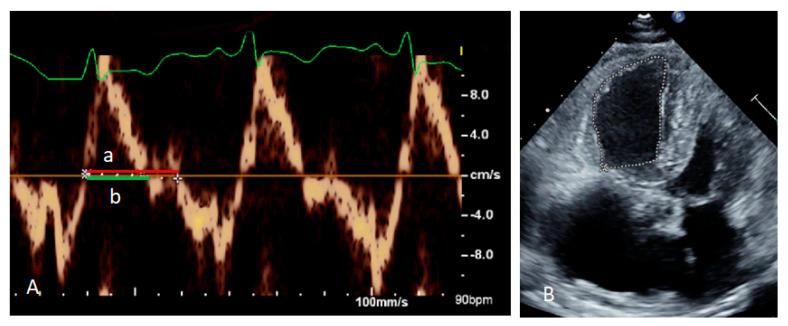
Right Ventricular Index of Myocardial Performance (RIMP) equals (a − b)/b, and the value was 0.64; “a” means tricuspid closure-open time, and “b” means ejection time, respectively. (**A**) The fractional area change of the right ventricle (RV FAC) on the four−chamber view was 21% (**B**).

**Figure 4 jpm-13-00062-f004:**
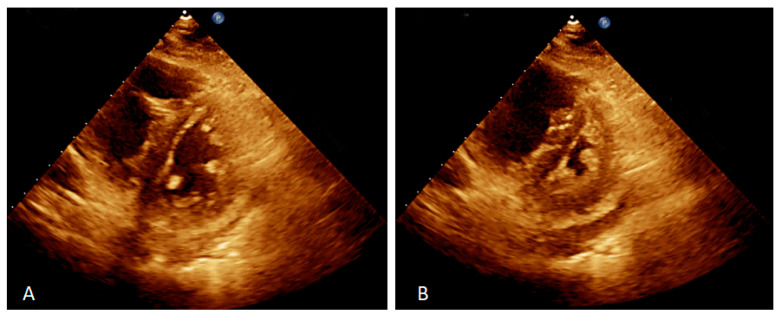
Two-dimensional echocardiography on the parasternal short-axis view showed D-shaped left ventricle during diastole, indicated for fluid overload (**A**) and systole, indicated for pressure overload (**B**).

**Table 1 jpm-13-00062-t001:** Laboratory data.

Item	Result
White blood cell	Normal
Hemoglobin	Normal
Platelet	Normal
Alanine aminotransferase	Normal
Aspartate aminotransferase	Normal
Blood urea nitrogen	27 mg/dL
Creatinine	1.42 mg/dL
High-sensitivity troponin I	48.8 pg/mL
N-terminal pro-brain natriuretic peptide	4528 pg/mL

**Table 2 jpm-13-00062-t002:** Echocardiographic results.

Item	Result
Left ventricular ejection fraction	59.7%
Systolic pulmonary artery pressure	96 mmHg
Right ventricle size	31 mm
RV index of myocardial performance	0.64
RV fractional area change	21%

RV—Right ventricle.

**Table 3 jpm-13-00062-t003:** Timeline of the patient’s treatments.

1989	● Diagnosis of ankylosing spondylitis. Patient started on sulfasalazine.
September 2021	● The patient was treated with diuretics.
October 2021	● 62.5 mg of bosentan twice per day was initiated for 7 days, and then the prescription was switched to 20 mg of sildenafil three times per day.
November 2021	● 10 mg of macitentan every day was added.

## Data Availability

Not applicable.

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
