# Peer review of "Case Report: A Rare Manifestation of Pulmonary Arterial Hypertension in Ankylosing Spondylitis"

_jpm, 2022, doi:10.3390/jpm13010062_

Round 1

Reviewer 1 Report

The authors present a case report on a rare manifestation of pulmonary arterial hypertension in ankylosing spondylitis”

Their study is based on the fact that there are very rare reports describing the link between AS and pulmonary arterial hypertension (PAH)

-The authors should improve the abstract and the conclusions because especially in the latter there are often repetitions

-From line 42 to 47, the authors present laboratory testing results. It would be appropriate to write in a clearer way , for example in a table.

 -From line 56 to 59, authors should arrange the data in a table for easier reference

-The authors could draw a timeline on which specify the patient's treatments

Author Response

The authors present a case report on a rare manifestation of pulmonary arterial hypertension in ankylosing spondylitis”

Their study is based on the fact that there are very rare reports describing the link between AS and pulmonary arterial hypertension (PAH)

-The authors should improve the abstract and the conclusions because especially in the latter there are often repetitions

ANS: Thank you for your comment! We have been revised it (Yellow part).

-From line 42 to 47, the authors present laboratory testing results. It would be appropriate to write in a clearer way , for example in a table.

ANS: Thank you for your comment! We have been made a table for laboratory testing results (Table 1).

 -From line 56 to 59, authors should arrange the data in a table for easier reference

ANS: Thank you for your comment! We have been made a table for echocardiographic results (Table 2).

-The authors could draw a timeline on which specify the patient's treatments

ANS: Thank you for your comment! We have been draw a timeline on which specify the patient's treatments (Table 3).

Reviewer 2 Report

Dear Authors,

Your work is quite an interesting case report. You reported an ankylosing spondylitis (AS) patient with pulmonary arterial hypertension (PAH). Physicians should be aware of this rare comorbidity in AS patients. Early screening is necessary for symptomatic patients using modalities, such as echocardiography. Further investigation is needed to disclose the association between AS and PAH.

Bigger note: there is one in total - is it worth posting single case ops? Perhaps it is necessary to communicate with other people taking care of patients with PAH and ask if they have noted similar dependencies?

Minor remarks: Please adjust the article strictly according to the editors' recommendations. You have the wrong font and text formatting.

Author Response

Your work is quite an interesting case report. You reported an ankylosing spondylitis (AS) patient with pulmonary arterial hypertension (PAH). Physicians should be aware of this rare comorbidity in AS patients. Early screening is necessary for symptomatic patients using modalities, such as echocardiography. Further investigation is needed to disclose the association between AS and PAH.

Bigger note: there is one in total - is it worth posting single case ops? Perhaps it is necessary to communicate with other people taking care of patients with PAH and ask if they have noted similar dependencies?

ANS: Thank you for your comment! To the best of our knowledge, this is the first case of AS associated with PAH 32 years after the AS diagnosis with a detailed clinical description. We are recommended that physicians should be aware of this rare comorbidity in AS patients. Therefore, it is necessary to communicate with other people taking care of patients with PAH and ask if they have noted similar dependencies in patient with AS.

Minor remarks: Please adjust the article strictly according to the editors' recommendations. You have the wrong font and text formatting.

ANS: Thank you for your comment! We have been revised it according to the editors' recommendations.
